# Vav1 and mutant K-Ras synergize in the early development of pancreatic ductal adenocarcinoma in mice

Yaser Salaymeh[1], Marganit Farago[1], Shulamit Sebban[1], Batel Shalom[1], Eli Pikarsky[2], Shulamit Katzav[1]

To explore the contribution of Vav1, a hematopoietic signal transducer, to pancreatic ductal adenocarcinoma (PDAC) development, we generated transgenic mouse lines expressing, Vav1, K-Ras$^{G12D}$, or both K-Ras$^{G12D}$ and Vav1 in pancreatic acinar cells. Co-expression of Vav1 and K-Ras$^{G12D}$ synergistically enhanced acinar-to-ductal metaplasia (ADM) formation, far exceeding the number of lesions developed in K-Ras$^{G12D}$ mice. Mice expressing only Vav1 did not develop ADM. Moreover, the incidence of PDAC in K-Ras$^{G12D}$/Vav1 was significantly higher than in K-Ras$^{G12D}$ mice. Discontinuing Vav1 expression in K-Ras$^{G12D}$/Vav1 mice elicited a marked regression of malignant lesions in the pancreas, demonstrating Vav1 is required for generation and maintenance of ADM. Rac1–GTP levels in the K-Ras$^{G12D}$/Vav1 mice pancreas clearly demonstrated an increase in Rac1 activity. Treatment of K-Ras$^{G12D}$ and K-Ras$^{G12D}$/Vav1 mice with azathioprine, an immune-suppressor drug which inhibits Vav1's activity as a GDP/GTP exchange factor, dramatically reduced the number of malignant lesions. These results suggest that Vav1 plays a role in the development of PDAC when co-expressed with K-Ras$^{G12D}$ via its activity as a GEF for Rac1GTPase.

## Introduction

Vav1, a signal transducer protein which is physiologically expressed in the hematopoietic system, was first identified as an in vitro–activated oncogene (Katzav et al, 1989). It functions as a GDP/GTP exchange factor (GEF) for Rho/RacGTPases, an activity that is stringently controlled by tyrosine phosphorylation (Crespo et al, 1997). This GEF activity of Vav1 regulates cytoskeletal rearrangement during immune cell activation (Fischer et al, 1998; Holsinger et al, 1998). Vav1 also participates in GEF-independent signaling pathways, including the JNK, ERK, NF-κB, and NFATc1 pathways, and associates with numerous adapter proteins such as Shc, NCK, SLP-76, Grb2, and Crk (Tybulewicz, 2005). Although the physiological activity of Vav1 is well understood, its contribution to human cancer is only starting to emerge. Several recent studies have indicated that mutations in various domains of the Vav1 protein are present in human cancers such as adult T-cell leukemia/lymphoma (Kataoka et al, 2015), lung adenocarcinoma and squamous cell carcinomas (Campbell et al, 2016), and peripheral T-cell lymphomas (Abate, da Silva-Almeida et al, 2017). In addition, numerous studies have reported the unexpected expression of Vav1, normally found only in the hematopoietic system, in a variety of human cancers, such as neuroblastoma (Hornstein et al, 2003), lung (Lazer et al, 2009), breast (Lane et al, 2008; Sebban et al, 2013; Du et al, 2014; Grassilli et al, 2014), ovarian (Wakahashi et al, 2013), prostate (Kniazev Iu et al, 2003), esophageal (Zhu et al, 2017), and brain tumors (Lindsey et al, 2014). Notably, Vav1 expression was also identified in more than 50% of 95 examined pancreatic ductal adenocarcinoma (PDAC) tumor specimens (Fernandez-Zapico et al, 2005), a finding that was validated by Huang et al (2016). Patients with Vav1-positive tumors had a worse prognosis than patients with Vav1-negative tumors (Fernandez-Zapico et al, 2005; Huang et al, 2016). Sequence analysis of Vav1 cDNA from pancreatic cancer cell lines and tumors confirmed their expression of intact wild-type (wt) Vav1 (Fernandez-Zapico et al, 2005). The aberrant expression of Vav1 in pancreatic cancer was attributed to epigenetic changes (Fernandez-Zapico et al, 2005; Huang et al, 2016). Furthermore, Vav1 RNAi was found to abolish neoplastic cellular proliferation of human pancreatic cancer cell lines both in vitro and in vivo, even in the presence of oncogenic K-Ras (Fernandez-Zapico et al, 2005). The accumulating data, thus, clearly point to an important role of ectopically expressed wtVav1 in pancreatic cancer (Fernandez-Zapico et al, 2005; Huang et al, 2016), possibly through its activity as a GEF that regulates cytoskeletal organization and/or through its activity as a signal transducer that can affect growth factor/cytokine production. To date, however, the mechanisms that mediate this protumorigenic role of Vav1 in pancreatic cancer and the stages during tumorigenesis, at which such mediation occurs, are unknown.

The earliest identifiable precursor lesion to PDAC is acinar-to-ductal metaplasia (ADM), which progresses to a series of neoplastic

[1]Department of Developmental Biology and Cancer Research, Institute for Medical Research Israel-Canada, The Hebrew University Hadassah Medical School, Jerusalem, Israel [2]The Lautenberg Center for Immunology and Cancer Research and Department of Pathology, Institute for Medical Research Israel-Canada, The Hebrew University Hadassah Medical School, Jerusalem, Israel

Correspondence: shulamitk@ekmd.huji.ac.il

precursor lesions known as pancreatic intraepithelial neoplasia (PanIN) (Morris et al, 2010; Aichler et al, 2012; Storz, 2017). The earliest and most frequent genetic alteration found in low-grade PanIN-1A lesions is mutant K-Ras, which is present in >90% of PDACs (Morris et al, 2010; Aichler et al, 2012; Kanda et al, 2012; Storz, 2017). Several groups have generated sophisticated somatic mouse models that faithfully recapitulate human pancreatic cancer pathogenesis and progression from ADM to PanIN and eventually to PDAC (Hingorani & Tuveson, 2003; Bardeesy et al, 2006; Guerra et al, 2007; Izeradjene et al, 2007). Expression of mutant K-Ras$^{G12D}$ or K-Ras$^{G12V}$ in the murine pancreas is sufficient to initiate the development of ADM followed by PanIN (Hingorani & Tuveson, 2003; Seidler et al, 2008; Morris et al, 2010; Guerra et al, 2011). However, the low frequency of spontaneous progression of precursor lesions to invasive PDAC suggests that additional genetic and/or epigenetic aberrations are required for disease progression, including inflammation and/or additional molecular insults (Morris et al, 2010). Other molecular components within the epithelium that drive ADM include epidermal growth factor receptor (EGFR) (Ardito et al, 2012), TGF-α (Song et al, 1999), SOX-9 (Kopp et al, 2012), MIST1 (Shi et al, 2009), KLF4 (Wei et al, 2016), Rac1 (Heid et al, 2011), and phosphoinositide-3-kinase (PI3K) (Hill et al, 2010). Development of PanINs and PDACs can be accelerated by introducing inactivating mutations in the tumor suppressor genes *Cdkn2a*, *Trp53*, or *Dpc4*/*Smad4*, all of which occur frequently in human lesions as they progress to invasive PDAC (Jones et al, 2008; Morris et al, 2010; BiankinWaddell et al, 2012) and by enhancing critical signaling pathways such as those of EGFR, Raf/Mek/Erk, PI3K/Pdk1/Akt, and Ral (Lim et al, 2005; Feldmann et al, 2010; Collisson et al, 2012; Eser et al, 2013).

Here, we report the involvement of Vav1 in pancreatic cancer development and growth by using a novel transgenic mouse model, in which Vav1 is specifically expressed in pancreatic acinar cells. Significantly, because mutant K-Ras is prevalent in PDAC, we also examined whether expression of transgenic wtVav1 (hereafter Vav1) together with mutant K-Ras enhances pancreatic malignancy. We did this by generating a novel transgenic mouse model that inducibly expresses both genes specifically in the pancreas. Our results clearly indicate that co-expression of Vav1 and K-Ras$^{G12D}$ dramatically increases the prevalence and decreases the time course required for malignant pancreatic lesions to appear, in comparison with the expression of K-Ras$^{G12D}$ alone, thus strongly suggesting that these two proteins synergize to enhance the development of pancreatic tumors. An important finding was the significant increase in ADM initiation observed when both proteins are present. Expression of Vav1 alone in the pancreas did not lead to development of any malignant lesions. This study adds an important layer in delineating the earliest events involved in ADM/PanIN formation and in revealing their mechanistic roles. Such knowledge can be expected to provide vital information on the molecular pathways that are instrumental in initiating PDAC.

# Results

### Generation of a Vav1 transgenic mouse line with inducible pancreatic-specific expression

To study the contribution of Vav1 to the development of pancreatic cancer tumors, we first generated a Vav1 transgenic mouse line. We

did this by using the Tet-On system, in which rtTA binds, only in the presence of Dox, to a tetO plasmid. For that purpose, we subcloned human Vav1 into a plasmid that encodes a tetO-responsive bidirectional promoter (tetO7minCMV) that drives expression of tetO–Vav1 hooked to GFP (Fig S1A). To validate the ability of our generated tetO–Vav1 plasmid to produce Vav1 upon activation, we transfected HEK293 cells with this plasmid and with a vector that encodes the rtTA. After treatment with Dox, the expression of Vav1 protein was observed, as verified by Western blotting (Fig S1B) and by immunofluorescence (Fig S1C). To validate the activity of the Vav1–tetO plasmid in vivo, we transfected it into mouse blastocysts by using an in vitro fertilization procedure to generate tetO–Vav1 mice. To ensure that these transgenic mice express Vav1 under the correct conditions, we crossed them with LAP–rtTA transgenic mice, which express rtTA in hepatocytes. As expected, Western blot analysis of tissue lysates from tetO–Vav1/LAP–tTA mice (positive mice, + in Fig S2) showed GFP expression (indicative of expression of the Vav1 transgene) in the liver but not in the spleen. LAP–rtTA transgenic mice lacking the Vav1 transgene (negative mice, Fig S2) did not express GFP in either of these tissues. Immunohistochemical analysis of livers from both positive and negative mice revealed expected staining pattern in hepatocytes (data not shown), thereby validating the tissue and cell type–specific expression of the Vav1 transgene.

To drive the expression of Vav1 in pancreatic acinar cells, we used a Ptf1a promoter, which participates in the maintenance of exocrine pancreas-specific gene expression (Krapp et al, 1998). We crossed Ptf1a–CreER mice with Lox-Stop-Lox (LSL)–rtTA and tetO–Vav1 mice to generate a Vav1 pancreatic mouse (Ptf1a–CreER/LSL–rtTA/tetO–Vav1; herein denoted Vav1 mouse).

The mutant K-Ras$^{G12D}$ is present in nearly 90% of human pancreatic cancers, and is, thus, sometimes co-expressed with endogenous Vav1. To investigate potential interactions between K-Ras$^{G12D}$ and Vav1 in PDAC development, we crossed the LSL–K-Ras$^{G12D}$ strain (which carries an LSL termination sequence bearing the K-Ras$^{G12D}$ point mutation) to the Ptf1a–CreER strain expressing Cre recombinase, under the control of Ptf1a. After treatment with tamoxifen, Cre recombination deletes the transcriptional termination sequence and allows K-Ras$^{G12D}$ to be expressed in the pancreas (Ptf1a–CreER; LSL–K-Ras$^{G12D}$; herein denoted K-Ras$^{G12D}$ mouse). The K-Ras$^{G12D}$ mouse was then crossed with the Vav1 mouse to generate a transgenic mouse line that expresses both Vav1 and K-Ras$^{G12D}$ in pancreatic acinar cells (Ptf1a–CreER; LSL–rtTA/LSL–K-Ras$^{G12D}$/tetO–Vav1; herein denoted K-Ras$^{G12D}$/Vav1 mouse). Mice expressing Ptf1a–CreER served as controls. At the age of 1 mo, all of the mice outlined were injected subcutaneously, twice over a 2-d interval, with 8 mg of tamoxifen. To ensure comparable experimental conditions, all of those mice were also treated with Dox, although this affects only Vav1 expression. Thus, whereas K-Ras$^{G12D}$ was constitutively expressed after elimination of the LSL sequence, Vav1 could be further regulated by either addition or discontinuation of Dox.

### Malignant lesions in the pancreas

Mice were euthanized at various time points (1, 2, 3.5, 5, and 12 mo) after transgene induction was started by tamoxifen and Dox

administration, and tissues removed from the pancreas, liver, lung, and spleen were analyzed for expression of the human Vav1 transgene as detected by GFP staining (Fig 1) and by anti-Vav1 Abs to detect endogenous Vav1 expression in K-Ras^G12D mice (Fig S3).

Staining for GFP (co-expressed with the Vav1 transgene) validated the expression of Vav1 in acinar pancreatic cells of Vav1 mice and in acinar and ductal pancreatic cells of K-Ras^G12D/Vav1 mice (Fig 1). No staining of GFP was noted in the liver, lung, and spleen (data not shown). We also surveyed the endogenous murine expression of Vav1 in the pancreas of K-Ras^G12D mice by staining with anti-Vav1 antibodies (Fig S3). Endogenous Vav1 expression was intensified with time in acinar and ductal cells after the start of K-Ras^G12D transgene induction, exhibiting an increase of ~150-fold after 12 mo of transgene induction compared with its expression after transgene induction for 1 mo (Fig S3B). Importantly, such endogenous Vav1 expression was frequently observed in the malignant lesions that appeared in these mice (Fig S3A).

We then analyzed the appearance of premalignant and malignant lesions, including ADM, PanINs, and PDAC (APPD), by hematoxylin and eosin (H&E) staining of sections at different times after the onset of transgene induction, as indicated in Fig 2. Appearance of lesions in the pancreas were observed already at 1 mo after transgene induction and onwards. Representative examples of pancreatic sections from the tested mouse lines 3.5-mo after transgene induction are depicted in Fig 2A. For each section, we calculated the degree of malignancy (the "transformation index," i.e., the APPD ratio expressed as APPD%), as described in the Materials and Methods section (Fig 2). APPDs were observed only in the pancreata of K-Ras^G12D and K-Ras^G12D/Vav1 mice (Fig 2). Remarkably, as seen in Fig 2B, at 3.5 mo and at 5 mo after the onset of transgene induction, co-expression of Vav1 and K-Ras^G12D dramatically increased the rate of development of pancreatic APPD, which progressed much faster than when either gene was expressed alone. This finding strongly suggested that mutant K-Ras and Vav1 synergize to enhance pancreatic tumor development (Fig 2B). We did not observe any malignant lesion in

tissues analyzed except the pancreas. Notably, when Vav1 was expressed alone, there was no evidence of APPD in the pancreas even when examined 12 mo after Vav1 transgene induction, indicating that expression of Vav1 by itself is not sufficient for tumorigenicity (Fig 2). An increase in APPD in K-Ras^G12D/Vav1 mice was already observed at 1- and 2-mo post-oncogene induction, yet apparently time is needed for further enhancement of APPD, as was seen in the later time points. After 12 mo of Vav1 transgene induction, the APPD ratio in K-Ras^G12D mice had caught up with that in K-Ras^G12D/Vav1 mice, and the previously observed difference in APPD ratio between them had disappeared. This might reflect an increase in endogenous Vav1 expression in the K-Ras^G12D mice, as indicated in Fig S3. At 5 mo after both transgenes were induced, we compared the APPD ratio obtained from K-Ras^G12D/Vav1 mouse pancreata after treatment with both tamoxifen and Dox to that obtained from K-Ras^G12D/Vav1 mouse pancreata treated with tamoxifen (which induces K-Ras^G12D expression) but not with Dox (which induces the expression of Vav1 transgene) (Fig S4). As shown in Fig S4A, K-Ras^G12D/Vav1 mice that did not receive Dox treatment exhibited substantially fewer APPDs than their Dox-treated counterparts. Quantified results of H&E staining from several mice clearly demonstrated a significant difference in APPD ratios between Dox-treated and Dox-untreated K-Ras^G12D/Vav1 mice (Fig S4B), suggesting that there was no leakiness in our experimental mouse system.

The synergistic effect of Vav1 and K-Ras^G12D on the development of malignancy was further highlighted by the significantly higher prevalence of PDAC in transgenic mice expressing both Vav1 and K-Ras^G12D than in mice expressing K-Ras^G12D only (Fig 3A and B). Whereas PDAC developed in K-Ras^G12D/Vav1 mice already at 1-mo post-transgene induction (1 case), K-Ras^G12D mice developed PDAC at a later stage post-oncogene induction (3.5- and 5-mo) at significant lower numbers (Fig 3B).

To ascertain that the synergism between K-Ras^G12D and Vav1 in APPD development stems from expression of the Vav1 transgene, Dox (which induces Vav1 transgene expression) that was administered to K-Ras^G12D/Vav1 mice (in which both transgenes had been

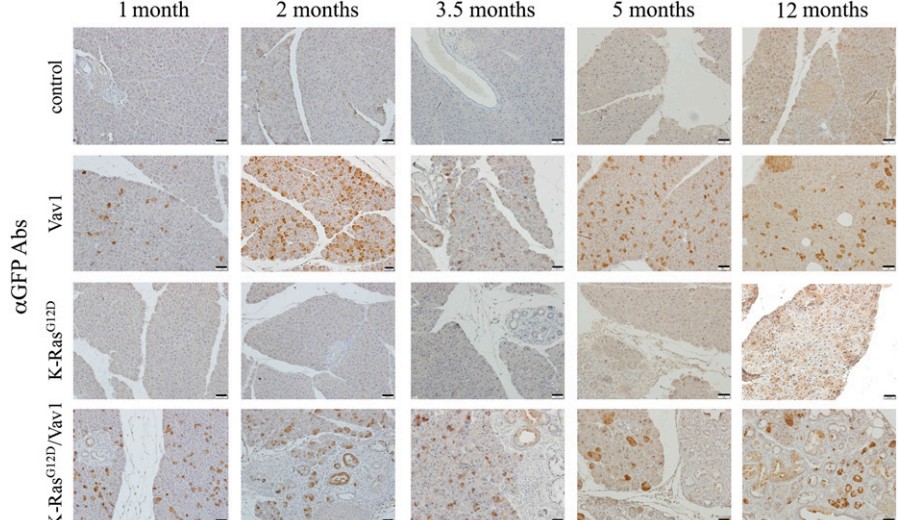

**Fig 1. Expression of GFP in the pancreata from different mouse lines.**
Representative paraffin sections of the pancreata of the various mouse lines at different time points after the onset of transgene induction (as indicated) were stained with anti-GFP Abs that identify the Vav1 transgene. Scale bar represents 25 μm. Number of mice stained in this experiment are outlined in Table S1.

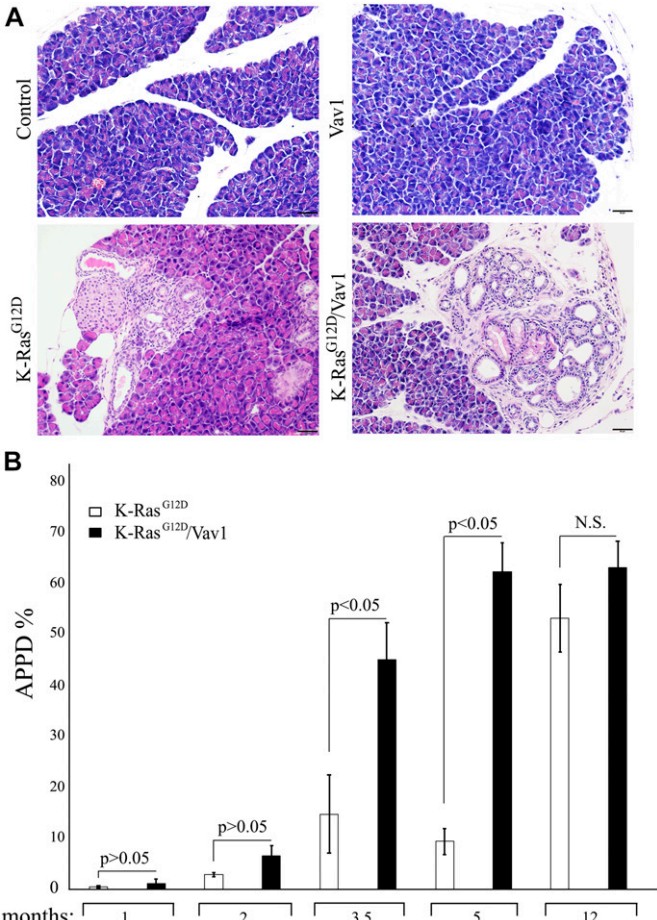

**Fig 2. Vav1 and K-Ras$^{G12D}$ synergize in the generation of malignant pancreatic lesions.**
Hematoxylin and eosin (H&E)–stained pancreata of control, Vav1, K-Ras$^{G12D}$, and K-Ras$^{G12D}$/Vav1 mice, at 1, 2, 3.5, 5, and 12 mo after the onset of transgene induction were analyzed for the appearance of malignant lesions. **(A)** Representative pictures of H&E–stained sections taken 3.5 mo after the onset of transgene induction are shown. Scale bar represents 20 μm. **(B)** The extent of malignant lesions generated in K-Ras$^{G12D}$ and K-Ras$^{G12D}$/Vav1 mice at the indicated time points after onset of transgene induction was calculated. The sum of area occupied by ADM, PanINs, and PDAC lesions (APPD) was measured as a fraction of the total area of the pancreas (APPD%). Numbers of K-Ras$^{G12D}$ and K-Ras$^{G12D}$/Vav1 mice used, respectively, were $n = 4$ and $n = 8$ at 1 mo; $n = 6$ and $n = 8$ at 2 mo; $n = 13$ and $n = 17$ at 3.5 mo; $n = 9$ and $n = 9$ at 5 mo; and $n = 7$ and $n = 5$ at 12 mo. Significant differences between the two analyzed groups ($P < 0.05$; $t$ test) are indicated. N.S. refers to statistical nonsignificant differences. SEM are shown.

induced for 3.5 mo) was stopped 20 d before analysis (Fig 3C and D). Staining of sections of the pancreas from K-Ras$^{G12D}$/Vav1 mice deprived of Dox clearly indicated a marked reduction in Vav1-positive cells, visualized by GFP staining (Fig 3D). The resulting depletion of transgenic Vav1 reduced the APPD ratio to the level observed in K-Ras$^{G12D}$ mice, that is, mice in which transgenic Vav1 was not expressed (Fig 3C and D). This finding indisputably suggests that continuous Vav1 expression is needed for maintenance of ADM lesions and/or their progression to cancer in K-Ras$^{G12D}$/Vav1 mice.

Our next step was to examine acinar and ductal cell compartments in the pancreata of control, Vav1, K-Ras$^{G12D}$, and K-Ras$^{G12D}$/Vav1 mice for differences in cell proliferation. This was done by

staining for Ki-67 protein, a marker for cellular proliferation. An increase in Ki-67 staining of acinar and ductal cell compartments was observed in K-Ras$^{G12D}$ and K-Ras$^{G12D}$/Vav1 mice (Fig 4A), with significant differences between these mice at 2, 3.5, and 5 mo after transgene induction (Fig 4B). No statistically significant differences in proliferation were observed between these two groups of mice at 1-mo and at 12-mo post-induction (Fig 4B). These results agree with the APPD scores observed in K-Ras$^{G12D}$ and K-Ras$^{G12D}$/Vav1 mice (Fig 2B). Furthermore, the fact that no differences in Ki-67 staining were observed between K-Ras$^{G12D}$ and K-Ras$^{G12D}$/Vav1 mice at the latest timepoint supports our claim that the lack of differences between these two groups is attributable to the increased expression of endogenous Vav1 (Fig S3).

### Signaling in transgenic mouse lines

The synergy observed here between Vav1 and K-Ras$^{G12D}$ in the development of pancreatic cancer could conceivably stem from signaling pathways in which both Vav1 and K-Ras$^{G12D}$ participate. To examine this possibility, we analyzed the status of EGFR and Erk activation in our control, Vav1, K-Ras$^{G12D}$, and K-Ras$^{G12D}$/Vav1 mice (Fig 5). EGFR and Erk phosphorylation were similarly increased in malignant pancreatic lesions both in K-Ras$^{G12D}$ and in K-Ras$^{G12D}$/Vav1 mice (Fig 5A and B). Western blot analysis of phospho-Erk and Erk in pancreatic tissues of the different mouse lines at the various time points after transgene further supported the immunohistochemistry results, indicating the absence of any synergistic effect on Erk phosphorylation when Vav1 and K-Ras$^{G12D}$ are co-expressed (Fig 5B and C). These results raised the possibility that the contribution of Vav1 to pancreatic cancer development stems from a signaling pathway that is unique to Vav1.

The best-known function of Vav1 is its tyrosine phosphorylation–dependent GEF activity for the Rho family of GTPases (Crespo et al, 1997). Several studies indicated that the activity of Vav1 as a GEF towards Rac1 plays an important role in Vav1's involvement in cancer (Fernandez-Zapico et al, 2005; Lazer et al, 2009). To determine whether activation of Rac1 might play a role in the contribution of Vav1 to development of APPDs, we analyzed Rac1 activation using specific anti–Rac1–GTP antibodies (Zhou et al, 2018) (Fig 6). We found that Rac1 activity was substantially more evident in pancreatic tissues of K-Ras$^{G12D}$/Vav1 than in control, Vav1, or K-Ras$^{G12D}$ pancreatic tissues (Fig 6A). Quantification of these results unequivocally demonstrated that Rac1 activation is synergistically increased in the K-Ras$^{G12D}$/Vav1 mouse line, where the two genes are co-expressed (Fig 6B). These results are also apparent when immunofluorescence with anti–Rac1–GTP Abs of pancreatic tissues from the mice used in our study were used (Fig 6C). Thus, although an increase in both EGFR and ERK phosphorylation was detectable in both K-Ras$^{G12D}$/Vav1 and K-Ras$^{G12D}$ mice, Rac1 activation was augmented only when Vav1 was ectopically co-expressed with mutant K-Ras. Notably, an increase in Rac1 activation in the K-Ras$^{G12D}$/Vav1 mouse pancreas was already evident 1 mo after transgene induction, a stage at which an increase in malignant lesions in these mice is only starting to emerge. Expression of Vav1 by itself was insufficient to cause an increase in Rac1 activation, probably because, as noted in Fig 5, there is no enough activating signals to lead to Vav1 tyrosine phosphorylation.

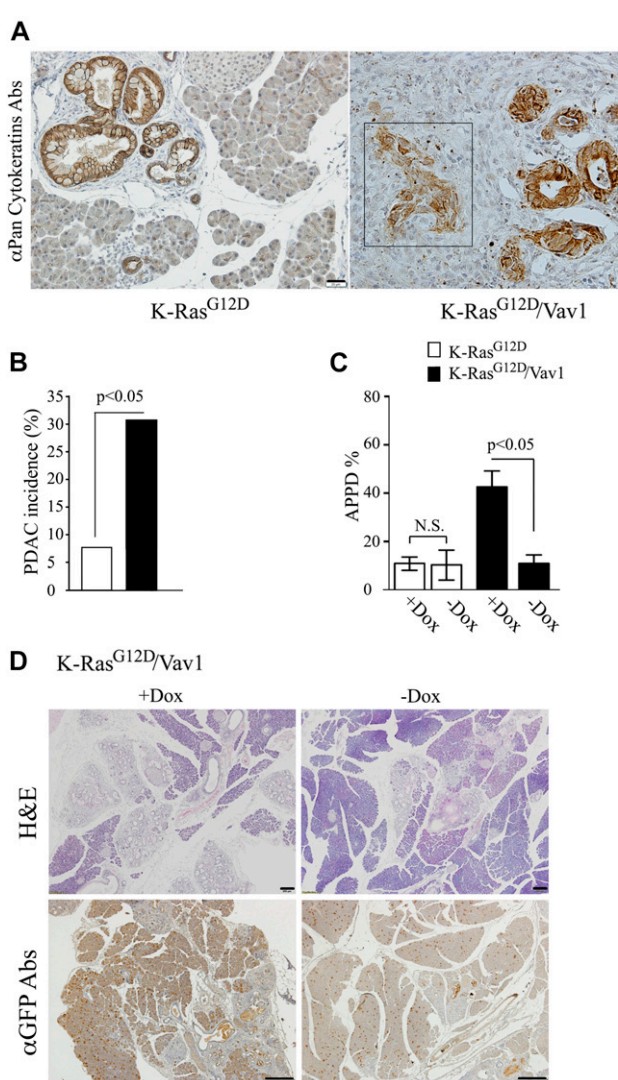

**Fig 3. Vav1 expression enhances pancreatic ductal adenocarcinoma (PDAC) generation in the K-Ras$^{G12D}$/Vav1 mouse pancreas and is critical for pancreatic malignant lesions.**

**(A)** Representative K-Ras$^{G12D}$ (left) and K-Ras$^{G12D}$/Vav1 (right) stained with anti–pan-cytokeratin Abs are shown. The presence of PDAC is shown in a representative pancreatic section from a K-Ras$^{G12D}$/Vav1 mouse 1 mo after transgene induction (square, right panel). Scale bar represents 20 μm. **(B)** The extent of PDAC present in K-Ras$^{G12D}$ and K-Ras$^{G12D}$/Vav1 mice at the different time points after transgene induction was assessed. The histogram summarizes the incidence of PDAC development in K-Ras$^{G12D}$ mice and in K-Ras$^{G12D}$/Vav1 mice. Two of 26 K-Ras$^{G12D}$ mice (7.7%) and 12 of 39 K-Ras$^{G12D}$/Vav1 mice (30.8%) had developed PDAC lesions. PDAC developed in K-Ras$^{G12D}$/Vav1 ranging from 1 to 12 mo post transgenes induction, whereas PDAC in K-Ras$^{G12D}$ developed in mice of 3.5 and 5 mo post transgene induction. Significance of the difference between them ($P < 0.05$) was calculated using a two-tailed chi-squared test (Fisher's exact test). **(C)** In some K-Ras$^{G12D}$/Vav1 mice (see numbers below), before completion of 3.5 mo of transgene induction, their Vav1 expression was discontinued for 20 d by removal of Dox from their drinking water (−Dox). Hematoxylin and eosin (H&E) staining of the pancreata of K-Ras$^{G12D}$ and K-Ras$^{G12D}$/Vav1 either treated with Dox and tamoxifen (+Dox) or in which Dox was removed from their drinking water for 20 d before completion of 3.5 mo of transgene induction (−Dox) was performed. All the mice were then analyzed for the appearance of malignant lesions. The number of malignant lesions generated in these mice was calculated as APPD%, both for those treated with Dox (+Dox; $n = 13$ and $n = 17$, respectively) and for those in which Dox was omitted for 20 d (−Dox; $n = 3$ and $n = 7$, respectively). Significant differences between the two analyzed groups ($P <$

The tight correlation between Vav1-induced Rac1 activity and development of APPD suggests that the activity of Vav1 as a GEF for Rac/RhoGTPases and not another cryptic activity of Vav1 is required for maintenance or further progression of pancreatic lesions.

### Decrease in APPD after treatment with azathioprine, an inhibitor of the Rac1 pathway

The results described above clearly point to a synergistic effect of Vav1 and K-Ras$^{G12D}$ inducing Rac1–GTP activation. To determine the extent of involvement of this pathway in the generation of APPDs in our mouse model, we used azathioprine (Aza). This drug is one of the oldest pharmacologic immunosuppressive agents used to treat hematologic malignancies, rheumatologic diseases, solid organ transplantation, and inflammatory bowel disease (Maltzman & Koretzky, 2003). Given its structure as a purine analog, it can become incorporated into DNA replication and can also block the de novo pathway of purine synthesis. Its activity as an immunosuppressive drug stems, however, from the specific blockade of Rac1 activation through binding of azathioprine-generated 6-thioguanine triphosphate (6-thio-GTP) to Rac1 instead of GTP. Such an activity prevents GEFs such as Vav1 from converting Rac1 to Rac1–GTP (Tiede et al, 2003). 2 mo after transgene activation, K-Ras$^{G12D}$ and K-Ras$^{G12D}$/Vav1 mice were injected i.p. with azathioprine (10 mg/kg) 5 d a week for 1 mo and were euthanized 15 d after the last injection (Fig 7). Remarkably, treatment with azathioprine significantly decreased APPD in the K-Ras$^{G12D}$/Vav1 mice, as seen in H&E–stained pancreatic sections of mice treated or untreated with azathioprine (Fig 7A). Calculation of APPD score in these experiments indicated that it is reduced from 45% in untreated K-Ras$^{G12D}$/Vav1 mice to 7% in Aza-treated mice (Fig 7B). We then tested whether treatment with azathioprine indeed leads to Rac1–GTP reduction. For that, we used Western blotting to analyze Rac1–GTP levels in pancreatic tissues from K-Ras$^{G12D}$ and K-Ras$^{G12D}$/Vav1 mice (Fig 7C and D). Our results clearly pointed to a marked reduction in Rac1–GTP levels in azathioprine-treated (+) K-Ras$^{G12D}$/Vav1 mouse pancreas versus non-treated mice (−), whereas no statistically significant differences were observed in the pancreata of K-Ras$^{G12D}$ mice treated similarly (Fig 7C). Quantification of these results revealed a ninefold decrease in Rac1 activation in azathioprine-treated K-Ras$^{G12D}$/Vav1 mice, whereas no such differences were observed in K-Ras$^{G12D}$ mice (Fig 7D). Thus, our results convincingly indicate that the activity of Vav1 as a GEF towards Rac1 is essential for Vav1's contribution to the development of malignant pancreatic lesions.

## Discussion

In the current study of the role of Vav1 in tumor development, our novel transgenic K-Ras$^{G12D}$/Vav1 mouse model was used for the first

---

0.05; $t$ test) are indicated. N.S. refers to statistical nonsignificant differences. SEM are shown. **(D)** Pancreatic sections from K-Ras$^{G12D}$/Vav1 mice after 3.5 mo post transgene induction either treated with Dox (+Dox) or in which Dox treatment was discontinued for 20 d (−Dox) were either stained by H&E (upper panel) or anti-GFP Abs (lower panel). Representative pictures are shown. Scale bar represents 200 μm.

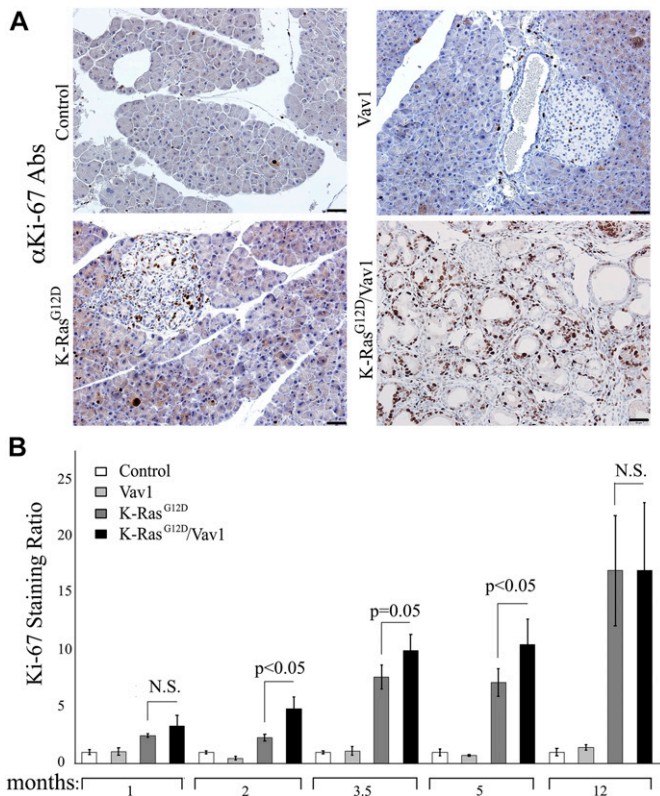

**Fig 4. Proliferation of pancreatic cells from control, Vav1, K-Ras[G12D], and K-Ras[G12D]/Vav1 mice.**

Proliferation of acinar and ductal pancreatic cells obtained, at different times after initiation of transgene induction, from control, Vav1, K-Ras[G12D], and K-Ras[G12D]/Vav1 mice was assessed by anti-Ki-67 staining. **(A)** Representative pictures of Ki-67–stained pancreatic sections taken from mice 5 mo after the start of transgene induction. Scale bar represents 50 $\mu$m. **(B)** Proliferation was quantified by counting 10 fields from each pancreatic section and calculating the mean value. The numbers of mice used for Ki-67 staining calculations are recorded in Table S1. To obtain the Ki-67 staining ratio in each case, the result at each time point was divided by the result obtained for the control at the same time point. Significant differences between the two analyzed groups ($P < 0.05$; $t$ test) are indicated. N.S. refers to statistical nonsignificant differences. SEM are shown.

time. We show that expression of Vav1 together with that of K-Ras[G12D] in the pancreas has a synergistic effect in enhancing ADM, which leads eventually to an increase in the number of PanINs. Thus, the number of malignant lesions observed in our mice co-expressing Vav1 and K-Ras[G12D] exceeded the number of lesions developed in mice that expressed only K-Ras[G12D]. This finding was accompanied by a significant increase in the incidence of PDAC in K-Ras[G12D]/Vav1 mice compared with that occurring in K-Ras[G12D] mice and by elevated proliferation rates as indicated by Ki67-positive acinar and ductal cells. When Vav1 was the only gene expressed, it did not lead to the development of any malignant lesions in the pancreas, suggesting that Vav1 can influence tumor development only when it functions together with other signaling proteins, such as K-Ras[G12D]. These results are in line with the observations that the survival rate of Vav1-positive human primary pancreatic adenocarcinomas was worse than that of Vav1-negative tumors (Fernandez-Zapico et al, 2005; Huang et al, 2016). Taken

together, our results point to an important contribution of Vav1, when co-expressed with K-Ras[G12D], to the development of ADM.

ADM of the pancreas is a process in which pancreatic acinar cells differentiate into cells with ductal cell traits. Data from genetic mouse models have shown that transgenic expression of oncogenic K-Ras[G12D] or K-Ras[G12V] in acinar cells initiates ADM and locks the cells into a transdifferentiated duct-like state (Hingorani & Tuveson, 2003; Guerra et al, 2007; Seidler et al, 2008; Morris et al, 2010; Pylayeva-Gupta et al, 2011). However, the low frequency of spontaneous development of ADM and its progression to invasive PDAC suggests that disease progression requires additional genetic and epigenetic aberrations, such as inflammation and/or additional molecular insults (Hingorani & Tuveson, 2003; Guerra et al, 2007; Seidler et al, 2008; Morris et al, 2010; Pylayeva-Gupta et al, 2011). Appearance of other mutations observed in human PDAC, including inactivation of the *P16INK4A/P19ARF, TRP53,* or *SMAD4* tumor suppressors as well as activation of the Hedgehog signaling pathway, significantly accelerates tumor development leading to acquisition of a metastatic phenotype (Hingorani et al, 2005; Bardeesy et al, 2006; Ijichi et al, 2006; Pasca di Magliano et al, 2006). In the present study, we found that Vav1 activation dramatically accelerates K-Ras–dependent ADM formation. Remarkably, discontinuing transgenic Vav1 expression (via removal of Dox from the drinking water of K-Ras[G12D]/Vav1 mice) in mice with extensive established ADMs led to a marked decrease in ADM lesions in the pancreas. This indicated that Vav1-dependent signals are necessary for the maintenance of the ADM state; upon loss of Vav1 expression, the ADM lesions rapidly regained their normal cell state. Our data reveals that ADM is a highly reversible lesion that depends on ongoing singaling through Rac1. This finding may have important therapeutic and/or preventive implications in pancreatic cancer.

Vav1 can potentially contribute to ADM development through its activity as a signal transducer. Our results demonstrate an increase in phosphorylation of EGFR and of Erk in pancreatic lesions of K-Ras[G12D]/Vav1 and K-Ras[G12D] mice. However, these results did not point to an enhanced synergistic effect of Erk activation in K-Ras[G12D]/Vav1 mice, as opposed to the finding in K-Ras[G12D] mice. The fact that there was no hint of synergism in Erk activation upon co-expression of K-Ras[G12D] and Vav1 indicates that the contribution of Vav1 might stem from its activity as a GEF towards Rac1. Analysis of the Rac1–GTP level in the pancreata from K-Ras[G12D]/Vav1 mice compared with that from K-Ras[G12D] or Vav1 mice indeed clearly demonstrated an increase in Rac1 activity, far exceeding the sum of its activities in the individual K-Ras[G12D]/Vav1 and K-Ras[G12D] mice. The well-known function of Vav1 as the tyrosine phosphorylation–dependent GEF activity for Rac1 (Crespo et al, 1997) was previously shown to be linked to an increase in tumorigenic properties of pancreatic cancer (Fernandez-Zapico et al, 2005; Lazer et al, 2010). Consistently with the notion that the activity of Vav1 as a GEF leads to its synergistic effect with K-Ras[G12D] on ADM generation, we observed an increase in Rac1 activity in K-Ras[G12D]/Vav1 mice, but not in Vav1 or in K-Ras[G12D] transgenic mice. Rac1 expression is increased in mouse and human pancreatic tumors, particularly in the stroma, and is generally a consequence of enhanced upstream inputs from receptor tyrosine kinases and phosphatidylinositol 3-kinases (PI3Ks) or of reduced Rac inactivation by GTPase-activating proteins (GAPs) (Heid et al, 2011). Deletion of Rac1 in the pancreata of K-Ras[G12D] mice reduces

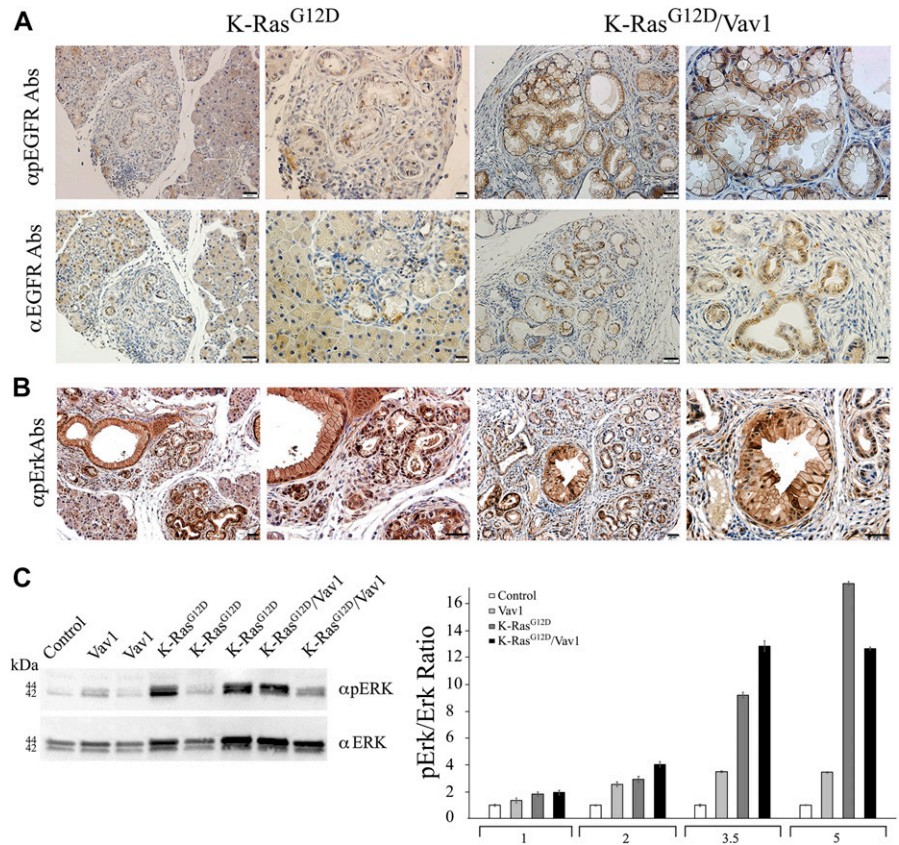

**Fig 5. Staining of phospho-EGFR and phospho-Erk in malignant pancreatic lesions from K-Ras^G12D/Vav1 mice.**

**(A)** Pancreatic sections from 4 K-Ras$^{G12D}$ and 8 K-Ras$^{G12D}$/Vav1 mice treated with tamoxifen and Dox for 5 mo were stained for pEGFR (upper panel) and total EGFR (lower panel). Representative pictures at two magnifications are shown. The scale bars represent either 50 $\mu$m (left pictures in each group) or 20 $\mu$m (right pictures in each group) as indicated. **(B)** Pancreatic sections from K-Ras$^{G12D}$ and K-Ras$^{G12D}$/Vav1 mice treated with tamoxifen and Dox for 5 mo were stained for pERK. Representative pictures are shown. Scale bar represents 20 $\mu$m. **(C)** Lysates from pancreatic tissues of 5-mo post-transgene induction from 3 control, 2 Vav1, 3 K-Ras$^{G12D}$, and 2 K-Ras$^{G12D}$/Vav1 mice were Western blotted using anti–phospho-Erk and anti-Erk (left panel). A representative blot from four independent experiments is shown. Relative Erk phosphorylation in the pancreata of control, Vav1, K-Ras$^{G12D}$, and K-Ras$^{G12D}$/Vav1 mice at 1, 2, 3.5, and 5 mo after the onset of transgene induction was calculated by quantifying their Western blots using ImageJ 1.49V software (right panel). Number of mice in these experiments are outlined in Table S1. SEM are shown.

ADM and consequently reduces PanIN and PDAC formation, leading to prolonged survival (Heid et al, 2011). Thus, in mice, Rac1 is required for early metaplastic changes and neoplasia-associated actin rearrangements in the development of pancreatic cancer (Heid et al, 2011).

Our results are further strengthened by the significant decrease recorded in K-Ras$^{G12D}$/Vav1 mice treated with the Rac1 blocker azathioprine. The azathioprine metabolite 6-thio-GTP specifically blocks the activation of Rac1. Its binding to Rac1 blocks Vav1's GEF activity upon 6-thio-GTP hydrolysis because of the accumulation of 6-thio-GDP–loaded Rac proteins, which Vav1 cannot convert to Rac–GTP (Tiede et al, 2003; Poppe et al, 2006). Furthermore, metastasis in a pancreatic cancer mouse model was shown to be inhibited with azathioprine through inhibition of the activity of Vav1 as a GEF towards Rac1 (Razidlo et al, 2015). Taken together, our results support the conclusion that the activity of Vav1 as a GEF for Rac1 is critical for the enhancement and maintaining of ADM generation.

This conclusion raises the question of why is Rac1 synergistically activated in K-Ras$^{G12D}$/Vav1 mice, but hardly active in mice that express either Vav1 alone or K-Ras$^{G12D}$ alone. We consider it possible that in the pancreas of the K-Ras$^{G12D}$/Vav1 mouse, there is an increase in chemokine/growth factor secretion that leads to heightened activation of Vav1. The microenvironment of pancreatic cancer contains many factors such as inflammatory cytokines and tumor associated macrophages, which influence the malignant status of the tumor. Such cytokines IL-8, IL-6, IL-1$\beta$, TNF-$\alpha$, IL-10, and others can be up-regulated and consequently contribute to tumor progression (Farajzadeh Valilou et al, 2018). Because Vav1 is a

participant in chemokine and cytokine signaling (Matsuguchi et al, 1995; Yuo et al, 1995; Dios-Esponera et al, 2015), it is conceivable that it could further amplify K-Ras and Vav1 activities in the pancreas through such pathways, thus potentially enhancing GM-CSF, CSF1, EGF, and/or TGF$\alpha$, which can function in an autocrine and/or a paracrine fashion. Our studies of lung cancer have suggested a potential crosstalk between cancer cells and the microenvironment, controlled by CSF1/Vav1 signaling pathways (Sebban et al, 2014), whereby Vav1 positively regulates the expression of CSF1, a growth factor that can lead to Vav1 activation in the microenvironment of tumor cells and in the immune cells in the microenvironment.

An additional potential possibility for the cooperation between Vav1 and mutant K-Ras involves the activity of c-Myc. Vav1 and mutant K-Ras co-operated in fibroblast transformation through overlapping downstream pathways that involve the transcription factor, c-Myc (Katzav et al, 1995). Also, Vav1 and c-Myc were shown to contribute together to the function of the immune system (Guy et al, 2013). c-Myc was found to enable transformation of embryo fibroblasts by a human Ras oncogene (Land et al, 1983). Several in vitro and in vivo data pointed to the contribution of c-Myc to mutant K-Ras pancreatic carcinogenesis (Skoudy et al, 2011). Furthermore, treatment of mutant K-Ras transgenic mice that developed PDAC with an anti-Myc drug showed an increase in cancer cell apoptosis, a reduction in cell proliferation, and a drastic attenuation of tumor growth, strongly suggesting anti-Myc drugs as potential chemotherapeutic agents for the treatment of PDAC (Stellas et al., 2014). Taken together, it is possible that c-Myc also participates as a

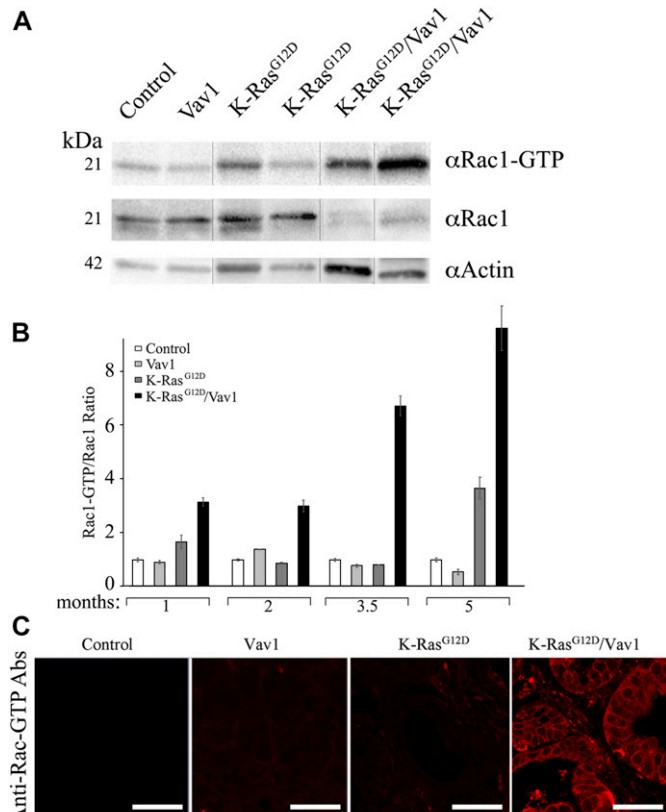

**Fig 6. Activation of Rac–GTP in malignant pancreatic lesions from K-Ras$^{G12}$/Vav1 mice.**

Rac1–GTP activation in the pancreata of control, Vav1, K-Ras$^{G12D}$, and K-Ras$^{G12D}$/Vav1 mice was analyzed. **(A)** Western blotting using anti–Rac1–GTP Abs, anti-Rac1, and anti-actin were used to evaluate Rac1–GTP levels in protein lysates from pancreatic tissues of control, Vav1, K-Ras$^{G12D}$, and K-Ras$^{G12D}$/Vav1 mice (two mice in each group) at 5 mo after transgene induction. The vertical black lines delineate sliced images that juxtapose lanes that were nonadjacent in the gel. **(B)** The levels of Rac1–GTP versus total Rac1 at the pancreata of control, Vav1, K-Ras$^{G12D}$, and K-Ras$^{G12D}$/Vav1 mice at 1, 2, 3.5, and 5 mo after initiation of transgene induction were calculated. Each Rac1–GTP/Rac1 was then divided by the control ratio. Band intensity of the Western blots was quantified using ImageJ 1.49V software. Number of mice in these experiments are outlined in Table S1. SEM are shown. **(C)** Immunofluorescence of Rac1–GTP of the pancreata from 2 control, 3 Vav1, 3 K-Ras$^{G12D}$, and 4 K-Ras$^{G12D}$/Vav1 mice, 3.5 mo post-transgene induction was performed. Representative pictures are depicted. Scale bar represents 25 μm. All sections were also stained with anti–Alexa Flour 594 dye for background analysis and found negative. Source data are available for this figure.

connecting downstream pathway in enhancing malignancy in K-Ras$^{G12D}$/Vav1 transgenic mouse model.

The findings of our study demonstrate that when Vav1 is co-expressed with mutant K-Ras, it participates as a GEF towards Rac1 in ADM generation. Taken together, our results thus point, for the first time, to the role played by Vav1 in the initial stages of pancreatic cancer development.

# Materials and Methods

## Mouse strains

Four mouse strains were used in this study: 1. Control mice—pancreas transcription factor 1 complex (Ptf1a)–CreER mice purchased from

Jackson Laboratories. 2. K-Ras$^{G12D}$ mice—generated by crossing Ptf1a–CreER mice with LSL–K-Ras$^{G12D}$ mice (Jackson Laboratories). 3. Vav1 mice—in which human Vav1 was subcloned into a plasmid encoding a tetO-responsive bidirectional promoter (tetO7minCMV), driving the expression of tetO–Vav1 hooked to GFP. TetO–Vav1–GFP mice were produced by microinjection of the plasmid into chimeric C57BL/6 mice blastocysts according to standard in vitro fertilization protocol. To ensure expression of Vav1 in pancreatic acinar cells, these mice were crossed with Ptf1a–CreER and LSL–rtTA mice. 4. K-Ras$^{G12D}$/Vav1 mice—produced by crossing of the K-Ras$^{G12D}$ mice with Vav1 mice to generate a mouse line having a Ptf1a–CreER; LSL–rtTA;LSL–K-Ras$^{G12D}$/tetO–Vav1. Details of the genotyping of these mice, including a detailed list of primers, are shown in Table S2.

To induce expression of the K-Ras$^{G12D}$ transgene in pancreatic acinar cells, we injected 1-mo-old mice subcutaneously with 8 mg of tamoxifen (2%; Sigma-Aldrich) dissolved in corn oil, twice over a 2-d interval. To induce Vav1 expression in these cells, doxycycline (Dox) (0.5 mg/ml; Bio Basic) was introduced into the drinking water of 1-mo-old mice as a sucrose (3% wt/vol) solution. To ensure comparable experimental conditions, all mouse strains used in this study were similarly treated with Dox and tamoxifen. All experiments were approved by the Hebrew University Ethics Committee for Animal Use (#MD-14-14199-5).

## Histology, immunohistochemical, and immunofluorescence analysis

Paraffin-embedded or frozen serial sections were subjected to H&E staining using standard procedures. Antibodies used for IHC are detailed in Table S3. Immunofluorescence of anti-Rac1 GTP was performed as follows: sections from paraffin-embedded blocks were used for immunofluorescence, deparaffinized using Xylene two time for 5 min each, followed by rehydration with decreased alcohol concentrations (100%–96%–80%), and then washed three times with PBS buffer. Antigen retrieval was performed by using citrate buffer in a pressure cooker. The sections were incubated for 1 h at room temperature with a blocking solution (CAS block; Invitrogen), followed by incubation with anti Rac1–GTP primary antibodies (1:200) overnight at 4°C in humidified chambers (Table S3). Slides were washed with PBS followed by incubation with antimouse Alexa Flour 594 secondary antibody. The slides were washed once with PBS for 5 min before nuclear staining with DAPI for 5 min and were dehydrated and mounted with coverslips. Pictures were taken using confocal microscopy (Nikon).

## Malignancy and proliferation measurements

H&E–stained sections from the pancreata of the mouse strains used in our study were quantified for malignant lesions. The degree of malignancy, or "transformation index," was calculated as the summed total area of ADM, PanINs, and PDAC lesions (APPDs) expressed as a percentage of the area of the entire pancreas (APPD %). PDAC cases identified in K-Ras$^{G12D}$/Vav1 and in K-Ras$^{G12D}$ pancreatic sections were counted, and their incidence was calculated in a blinded manner by a certified histopathologist. Proliferation rates were calculated by counting of positive Ki-67 acinar and ductal cells from the various mouse lines at different time

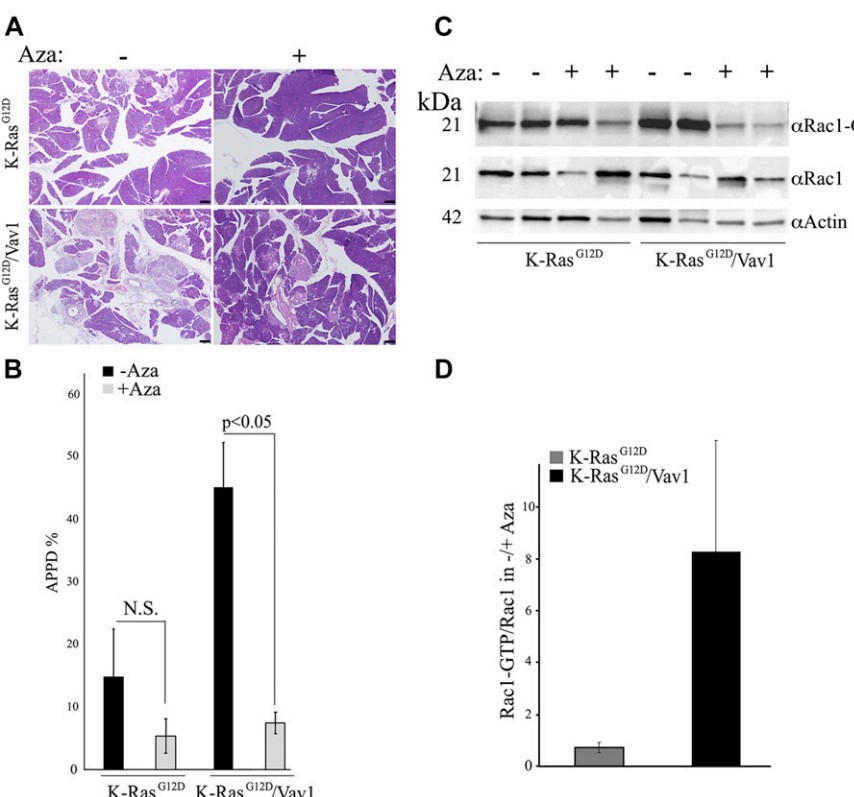

**Fig 7.  Treatment of K-Ras^G12D and K-Ras^G12D/Vav1 mice with azathioprine.**
At 2 mo after initiation of transgene induction, K-Ras$^{G12D}$ and K-Ras$^{G12D}$/Vav1 mice were injected i.p. with azathioprine (Aza; 10 mg/kg), 5 d a week for 1 mo, and euthanized 15 d after the last injection. **(A)** Representative pictures of hematoxylin and eosin (H&E)–stained sections of pancreata from Aza-untreated (indicated by –) and Aza-treated mice (indicated by +) are shown. Scale bar represents 200 μm. **(B)** The histogram shows the extent of malignant lesions (APPD%) generated in Aza-treated K-Ras$^{G12D}$ and K-Ras$^{G12D}$/Vav1 mice. Numbers of K-Ras$^{G12D}$ and K-Ras$^{G12D}$/Vav1 mice used without treatment (–) were $n = 8$ and $n = 12$, respectively, and numbers treated with Aza (+) were $n = 6$ and $n = 7$, respectively. Significant differences between the two analyzed groups ($P <$ 0.05; $t$ test) are indicated. N.S. refers to statistical nonsignificant differences. SEM are shown. **(C)** Western blotting using anti–Rac1–GTP Abs, anti-Rac1 and anti-actin were used to evaluate Rac1–GTP levels in protein lysates from pancreatic tissues of Aza-untreated (–) or Aza-treated (+) K-Ras$^{G12D}$ and K-Ras$^{G12D}$/Vav1 mice. **(D)** The relative ratio of Rac1–GTP/Rac1 level was calculated for each group of Aza-untreated (–) or Aza-treated (+) mice. Band intensity of the Western blots was quantified using ImageJ 1.49V software. SEM are shown.

points after transgene induction. Ten randomly chosen fields (at 400× magnification) from each pancreatic section were used for counting using ImageJ 1.49Vi.

### Western blotting

Pancreatic tissues were lysed in a lysis buffer containing 50 mM Tris–HCl, 150 mM NaCl, 25 mM EDTA, 1% NP-40, 1× phosphatase and protease inhibitors (Roche), and 1% PMSF. Tissues were fragmented using a homogenizer (Polytron Kinematica). Lysed tissues were incubated for 30 min on ice and then centrifuged at maximum speed for 20 min. Lysed proteins were quantified using Bio-Rad protein assay (Bio-Rad). Equal amounts of proteins (30 μg) were loaded on 4–15% gradient acrylamide gels, as previously described (Lazer et al, 2010). Antibodies used for Western blotting are listed in Table S3. Western blot results were quantified using ImageJ 1.49 V1.

### Azathioprine treatment

Azathioprine was diluted in PBS to a final concentration of 2 mg/ml and injected i.p. (10 mg/kg) into mice, 2 mo after transgene induction, 5 d a week for 1 mo. Mice were euthanized 15 d after the last injection.

### Statistical analysis

PDAC incidence was analyzed by the two-tailed Fisher's test. Significant differences (indicated by $P < 0.05$) were calculated according to Student's two-tailed test using the GraphPad Prism or Excel software.

## Supplementary Information

## Acknowledgements

We are indebted to Shirley Smith for editing the manuscript and to Dr Dror Gal, Prof. Ittai Ben-Porath, and Prof. Yuval Dor for advice on mouse manipulation. This work was supported, in part, by grants from the Israel Academy of Sciences, the Israel Cancer Association (ICA) with the generous assistance of the USA Friends of ICA in Honor of Laurence Holzman and Felicia Needleman; The Israel Cancer Research Fund; The Hubert H Humphrey Center for Experimental Medicine and Cancer Research; The Alex U Soyka Pancreatic Cancer Research Project; and a donation from Vibeke Lichten.

### Author's Contributions

Y Salaymeh: formal analysis, investigation, and methodology.
M Farago: investigation, methodology, project administration, and writing—original draft, review, and editing.
S Sebban: methodology.
B Shalom: formal analysis and methodology.

E Pikarsky: conceptualization and writing—original draft, review, and editing.

S Katzav: conceptualization, formal analysis, supervision, funding acquisition, and writing—original draft, review, and editing.

## Conflict of Interest Statement

The authors declare that they have no conflict of interest.

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
