## [Reviewer comments · Life Science Alliance]

Life Science Alliance

Vav1 and Mutant K-Ras Synergize in the Early Development of Pancreatic Ductal Adenocarcinoma in Mice

Yaser Salaymeh, Marganit Farago, Shulamit Sebban, Batel Shalom, Eli Pikarsky, and Shulamit Katzav

DOI: <https://doi.org/10.26508/lsa.202000661>

Corresponding author(s): Shulamit Katzav, Faculty of Medicine-The Hebrew University

Review Timeline:

Submission Date:	2020-01-28
Editorial Decision:	2020-02-19
Revision Received:	2020-03-16
Editorial Decision:	2020-03-18
Revision Received:	2020-03-20
Accepted:	2020-03-23

Scientific Editor: Andrea Leibfried

Transaction Report:

February 19, 2020

Re: Life Science Alliance manuscript #LSA-2020-00661-T

Prof. Shulamit Katzav
Faculty of Medicine-The Hebrew University
Developmental Biology & Cancer Research
P.O.Box 12272
Jerusalem 91120
Israel

Dear Dr. Katzav,

Thank you for submitting your manuscript entitled "Vav1 and Oncogenic K-Ras Synergize in the Early Development of Pancreatic Ductal Adenocarcinoma in Mice" to Life Science Alliance. The manuscript was assessed by expert reviewers, whose comments are appended to this letter.

As you will see, the reviewers appreciate your data and they provide constructive input on how to further strengthen your work. We would thus like to invite you to submit a revised version of your manuscript to us, addressing the individual points raised by the reviewers. This seems rather straightforward, but please do get in touch in case you would like to discuss specific revision points further with us.

Thank you for this interesting contribution to Life Science Alliance. We are looking forward to receiving your revised manuscript.

Sincerely,

B. MANUSCRIPT ORGANIZATION AND FORMATTING:

Reviewer #1 (Comments to the Authors (Required)):

In the present study, the authors have studied the role of Vav1 in PDAC development. The study establishes convincingly that Vav1 alone does not promote increased tumorigenicity, however it enhances the tumor development in KRASG12D mutated mice in a Rac1 dependent manner. The study reports important findings; but certain points needs to be addressed before considering the

Major comments:

1. In Fig1, in the first time point, GFP expression is detected but is this before the first injection of Tamoxifen or after? If it is before the injection, then, is represent leaky expression of Vav1? Furthermore, the authors claim that ducts are expressing GFP in Vav1 mice ("staining for GFP ... validated the expression of Vav1 in acinar and ductal pancreatic cells of both the Vav1 and the K-RasG12D/Vav1"), but it is not clear in the images that ducts are stained in Vav1 mice. Please provide images showing stained ducts.
2. In Figure 2B, the difference in APPD is only seen between 3.5 and 5 months, in the other time points the difference is not significance. Why is there no difference in later or earlier time points? Moreover in Figure 2, barely no APPD is present after 1 month, but in fig 3A the presence of PDAC is shown already after one month, how do authors explain these differences?
3. In Figure 4, only time point 5 is showing significant increase in ki-67 staining, other time points are not significant. How can authors be so sure about the association of Vav1 with proliferation when only one time point is significant. Considering that total tumor area is significantly higher in KRAS/Vav1 at time point 3.5 month, why is there no significant difference ki67 expression at that time point? What are the authors explanation for this? Please elaborate.
4. Is there any clinical relevance of high Vav1 expression in patients with PDAC? The manuscript would gain in scientific impact if the authors could provide some evidence of Vav1 beeing important for the human disease. For instance, is there any correlations between Vav1 expression and survival or treatment response (looking into publicly available data sets such as TCGA etc.)?

Minor comments:

5. In the method section for western blots, the authors have said that equal volume of protein was loaded on the gel, however, it is not mentioned how the amount of protein was quantified.
6. In Figure 1, image of Vav1 2 months, scale bar is missing. And for all images in Figure 1, the scale bars are to tiny to see.
7. In figure 2B, the APPD % of KRAS has decreased in month 5 compared to month 3.5. How do authors explain this difference?
8. There is a typo mistake in legend of Fig 3, it should be KrasG12D instead of KrasG12D/Vav1
9. In Fig5 B and C, the size of the error bars of the quantification are not corresponding with the variation of intensity seen in the western blot images. Can authors provide explanation for that?
10. Fig6B, the authors should provide the error bars for entire graph. Further, can authors provide the exact number of mice used instead of stating "several mice"?
11. Fig 7D, error bars are absent from the graph.
12. In the last section of the Results (the text relating to Fig C&D), the data is referred to twice in the text, which is redundant.
13. In the Discussion section, there is a typo in last but one paragraph. It should be "raise" instead of "rase".

Salaymeh and colleagues demonstrate a role for Vav1 in early development of pancreatic cancer. Vav1 is a GTP/GDP exchange factor that participates in signal transduction downstream of RTK signaling and is known for its roles in blood cells. The authors now show that its expression in pancreatic cancer has an interesting function: it acts together with the key PDAC oncogene, mutant Kras, to drive the first step of PDAC formation namely acinar to ductal metaplasia; in addition, their results suggest that vav1 is important also for the maintenance of PDAC. Strikingly, vav1 only synergizes with Kras, without causing ADM when expressed alone.

To reach these conclusions they use state of the art mouse models, combining acinar-specific expression of mutant Kras with acinar-specific dox-dependent expression of vav1, and using also biochemical GEF assays and a pharmacologic approach to inhibit vav1.

Overall this is an important paper that reveals an important role for vav1 in critical stages of PDAC development. The results are impressive and clean, and the interpretations are solid. The message of the paper gets through and is convincing despite the obvious weakness of using a gain of function approach.

I strongly support publication in LSA pending a revision that addresses the following concerns.

1. The authors state that "The removal of doxycycline from the drinking water of K-RasG12D/Vav1 mice (leading to a decrease in Vav1 transgene expression) elicited a marked decline in malignant lesions in the pancreas, further demonstrating that ADM generation requires the expression of Vav1 together with K-RasG12D". There is a conceptual flaw here: if vav1 was already expressed and gave rise to metaplastic lesions, decreasing its expression does not indicate vav1 requirement for ADM generation, but rather that vav1 is required for maintenance of ADM lesions and/or their progression to cancer. Please clarify. This is discussed to some extent already in the discussion; abstract has to reflect these points as well.

2. Figure 7 shows the use of Azathioprine as a pharmacologic approach, though not entirely specific, to inhibit vav1 activity and demonstrate a phenotype of fewer lesions. Please clarify in abstract the model in which this was done- kras;vav1, not kras only mice. The implication is that vav1 activity (and not another cryptic activity of vav1) is needed for maintenance or further progression of lesions developed due to excess vav1 activity.

3. Abstract lacks a concluding sentence- what is the bottom line regarding the role of vav1 in pdac?

4. If vav1 is required for ADM, it should be expressed in acinar cells, not just in full blown PDAC. Can the authors show this, in mouse and (preferably) in human material? Alternatively its expression in acinar cells undergoing metaplasia can be transient - please either show data or discuss in text.

5. Is there evidence on whether the effects of vav1 are cell autonomous (as would be the null hypothesis) or whether non-transgenic cells are also affected? The idea that cytokines turn the cell autonomous nature of vav1 activity into a microenvironmental effect is very interesting.

Reviewer #3 (Comments to the Authors (Required)):

In this paper, Salaymeh et al explore the contribution of Vav1 to onset and/or progression of pancreatic ductal adenocarcinoma. To this end, they developed transgenic mouse lines specifically expressing Vav1 and K-RasG12D, individually or combined, in pancreatic acinar cells. Based on characterization of the tumor lesions developing in their different mouse strains as well as on assays of activation of various signaling molecules, particularly Rac1, the authors conclude that Vav1 plays a significant synergistic role along KRAS G12D oncogenes during initial stages of pancreatic cancer development.

This manuscript provides new experimental tools and adds novel information/knowledge to this particular field. The data presented are clear and conclusive and the article is well written. However,

in my opinion, several issues should be addressed or clarified before this paper is considered acceptable for publication.

Comments

1) The authors use frequently the term "oncogene" throughout the manuscript to mention/describe the Vav1 gene used in this report. To avoid confusion, since the constructs used here contain specifically the WT version of Vav1, they should correct this language.

2) Figure 3: The authors need to provide stronger direct evidence to support/strengthen the conclusions they draw (page 9) from the current figure. I suggest, in particular, to (i) add WB assays showing the actual reduction of Vav1 under the conditions used (Dox removal 20 days prior to analysis), or (ii) at least, provide information on the half-life of the concerned cellular Vav1 proteins. Including a control panel of single K-RasG12D pancreas stained with anti-cytokeratine would also be advisable. Addressing these issues will also reinforce the conclusions drawn by the authors from the data in Figure 4.

3) Figure 5: A graph quantifying the pEGFR immunohistochemical signals detected in the different experimental groups (or direct WB data) would also be advisable here.

4) Figure 6: The data shown in the three panels of this figure clearly support the synergism between Vav1 and K-ras resulting in significant increase of the levels of activated Rac1-GTP as indicated in page 11. However, the authors need to clarify experimental details of the data presented:

- The Figure legend does not specify whether or not similar amounts of total protein were loaded in the WB assays in Fig 6 panel A. Is there a reduction of total Rac1 protein expressed upon co-expression of Vav1 and K-ras ? If so, the authors should mention this in the text. Similar questions arise when considering Figure 7: does azathioprine treatment cause increased expression levels of total Rac1 in comparison to untreated samples?
- The experimental details of the methodologies used for assays of Rac1 activation should be specified in Material and Methods. The authors seem to base all their determinations exclusively on WB assays using antibodies against Ras1-GTP, however other authors use immunoprecipitation and subsequent WB. The positive and negative controls for the immunofluorescence assays shown in panel C should also be shown, or at least mentioned.

5) Regarding mechanistic aspects:

- The authors suggest (p. 16) the possibility that Vav1 synergizes with oncogenic K-ras through its participation in chemokine and cytokine signaling. This view could be reinforced if the authors could include in this paper some data regarding the status of some inflammation biomarkers in the different experimental groups analyzed here.
- A Pubmed search on this subject and authors brings out an old publication (Katzav et al, Oncogene, 1995) entitled "Vav and Ras induce fibroblast transformation by overlapping signaling pathways which require c-myc function". In view of the many publications on pancreatic cancer and K-ras-mediated transformation that have been published in more recent years, the authors may consider discussing here whether the synergism proposed for Vav1 and K-ras may also require c-myc contribution.

Minor points

1) Please define units for panel B Figure EV 3B

2) Please correct typos (i.e: co-epxressed, p.8; imunnohistochemistry, p.10; singaling, p.14;

Remarkably, p.12; rases, p.16; ...)

3) Add parameters of statistical significance missing in some graphs (i.e.: Figure 5).

4) Number of mice used for each experiment must be indicated in all figure legends.

General Corrections:

- The abstract was rewritten based on the remarks of Reviewer #2
- Summary blurb was added
- Changes to the text, figure legends and Material & Methods as requested by the reviewers and specified below.
- List of changes made to Figures:
Figure 1- The image of Vav1 2-months post induction with the scale bar was added to the revised figure.

Figure 3-

- > IHC of K-Ras^{G12D} with anti-Cytokeratin was added, panel A, left panel.
- > The PDAC in K-Ras^{G12D}/Vav1 mice stained by anti-Cytokeratin Abs is highlighted by a black square, panel A, right panel.
- > A representative picture of K-Ras^{G12D}/Vav1 with Dox stained with and without GFP is added as part of panel D.

Figure 5-

- > Staining of sections of K-Ras^{G12D} and K-Ras^{G12D}/Vav1 mice with anti-EGFR abs is shown in A below the picture of stained sections with pEGFR Abs.
- > Pictures stained with pERK are now presented in B.
- > Panel C refers to western blots of pERK and ERK (left panel), while right panel represents the quantification of the ERK results.
- > Error bars were added.

Figure 6-

- > The level of Actin was added to panel A.
- > Error bars were added.

Figure 7-

- > The level of Actin was added to panel C.
- > Error bars were added to panel D.

Supplementary Figures- all figures are now referred to as S instead of EV.

Figure S3- The Y axis was changed to No. Vav1 positive cells.

Figure S4. The scale bars are detailed at the figure legend and therefore the magnification was removed from the figure.

Supplementary Tables

Table S2- the antibody used in the immunofluorescence of Rac-GTP (Figure 6) was added.

Table S3- Number of mice used in experiments presented in figures 1, 5, and 6 are documented. The number of mice used in the Ki-67 experiment (Fig. 4) was already part of this table.

We are now submitting a revised version of our manuscript in which we addressed all the reviewers' critiques, as follows:

Reviewer #1:

Major comments:

1. **Remark**- In Fig1, in the first time point, GFP expression is detected but is this before the first injection of Tamoxifen or after? If it is before the injection, then, is represent leaky expression of Vav1? Furthermore, the authors claim that ducts are expressing GFP in Vav1 mice ("staining for GFP ... validated the expression of Vav1 in acinar and ductal pancreatic cells of both the Vav1 and the K-RasG12D/Vav1"), but it is not clear in the images that ducts are stained in Vav1 mice. Please provide images showing stained ducts.

Response-

- All the time points indicated in the figure legends and the text refer to months after Tamoxifen and Dox treatment (transgenes induction), as indicated in the text, figure legends and Material and Methods. Tamoxifen was injected to one-month old mice, thus there is no leaky expression of Vav1. For example, **page 7, line 11**.
- The reviewer is correct and there is indeed no Vav1 expression in the ducts in Vav1 mice. The text was rewritten accordingly. Please see: **page 7, line 25**, as follows: "Staining for GFP (co-expressed with the Vav1 transgene) validated the expression of Vav1 in acinar pancreatic cells of Vav1 mice and in acinar and ductal pancreatic cells of K-Ras^{G12D}/Vav1 mice".

2. **Remark**- In Figure 2B, the difference in APPD is only seen between 3.5 and 5 months, in the other time points the difference is not significant. Why is there no difference in later or earlier time points? Moreover, in Figure 2, barely no APPD is present after 1 month, but in fig 3A the presence of PDAC is shown already after one month, how do authors explain these differences?

Response-

- We draw the attention of Reviewer #1 to the text "After 12 months of Vav1 transgene induction the APPD ratio in K-Ras^{G12D} mice had caught up with that in K-Ras^{G12D}/Vav1 mice, and the previously observed difference in APPD ratio between them had disappeared. This might reflect an increase in endogenous Vav1 expression in the K-Ras^{G12D} mice, as indicated in Fig. S3", **page 8, lanes 25 & 26, page 8, lanes 1-3**. This section explains why we do not see any differences in APPD at 12 months post transgene expression. As for earlier time points, 1 and 2 months, we conclude that there are accumulating events that need more time to occur for APPD to develop, **page 8, lanes 23-25**.

- Although, there are barely APPD after one month, one PDAC developed in K-Ras^{G12D}/Vav1 a month post-transgene induction (Figure 3A, now right panel). The PDACs counted represent the total number in all mice 1-month to 12-months post-transgene induction. There were **only** 2 PDAC cases in K-Ras^{G12D} mice at 3.5- and 5-months post oncogene induction. The number and timing of cases of PDAC cases in K-Ras^{G12D}/Vav1 mice is as follows: 1-month-2 PDACs; 2-months- 3 PDACs; 3.5-months-7 PDACs; 5-months-6 PDACs and 12-month-4 PDACs.

3. **Remark-** In Figure 4, only time point 5 is showing significant increase in ki-67 staining, other time points are not significant. How can authors be so sure about the association of Vav1 with proliferation when only one time point is significant. Considering that total tumor area is significantly higher in KRAS/Vav1 at time point 3.5 month, why is there no significant difference ki67 expression at that time point? What are the authors explanation for this? Please elaborate.

Response-

- Figure 4B shows significant differences at 2-months ($p < 0.05$), 3.5- months ($p = 0.05$) and 5-months ($p < 0.05$), post-transgene induction. Time points 1-month and time point 12-months, post transgene induction, are non-significant. I apologize for how we formulated the sentences relating to this issue in the previous version of the manuscript. We now corrected the wording, **page 10, lanes 8-10**. There are no changes in the figure as it reflects our results as explained here and in the revised manuscript, **page 10, lanes 8-10**.

4. **Remark-** Is there any clinical relevance of high Vav1 expression in patients with PDAC? The manuscript would gain in scientific impact if the authors could provide some evidence of Vav1 being important for the human disease. For instance, is there any correlations between Vav1 expression and survival or treatment response (looking into publicly available data sets such as TCGA etc.)?

Response-

- The association between Vav1 expression and pancreatic cancer has been discussed in the literature. Fernandez-Zapico ME, et al., analyzed 95 pancreatic cancer patients for the expression of VAV1 and concluded that 50% of cancers were positive and that “ VAV1-positive tumors had a worse survival rate compared to VAV1-negative tumors” (Fernandez-Zapico ME¹, Gonzalez-Paz NC, Weiss E, Savoy DN, Molina JR, Fonseca R, Smyrk TC, Chari ST, Urrutia R, Billadeau DD. *Cancer Cell*. 2005 Jan;7(1):39-49). **Page 13, lines 16-18.**

- Also, Huang PH et al., analyzed an array of 200 pancreatic cancer tissue samples from 94 patients and concluded that “the subgroup of PDAC patients with high VAV1 protein expression was significantly associated with a worse overall survival (32.4 and 11.8 months for patients with low (VAV1 low) and high (VAV1 high) VAV1 staining, respectively ($P < 0.001$) and a shorter time to cancer recurrence (16.7 versus 5.9 months)” (Huang PH, Lu PJ, Ding LY, Chu PC, Hsu WY, Chen CS, Tsao CC, Chen BH, Lee CT, Shan YS, Chen CS. *Oncogene*. 2017 Apr 20;36(16):2202-2214). **Page 13, lines 16-18.**

- These references were discussed and cited by us-please see **page 13, lines 16-18.**

Minor comments:

5. **Remark-** In the method section for western blots, the authors have said that equal volume of protein was loaded on the gel, however, it is not mentioned how the amount of protein was quantified.

Response-

- We loaded an equal quantity of protein and not volume. It is now described appropriately in the Material and Method section, see **page 19, lines 10, 11.**

6. **Remark-** In Figure 1, image of Vav1 2 months, scale bar is missing. And for all images in Figure 1, the scale bars are too tiny to see.

Response-

- The figure with the scale bar is added in the corrected figure. The scale bars are indicated in the legends of all figures as required by the journal.

7. **Remark-** In figure 2B, the APPD % of KRAS has decreased in month 5 compared to month 3.5. How do authors explain this difference?

Response-

- The standard error between these two groups is similar and there is no statistical difference between K-Ras 3.5-months to 5-months, it is 0.76.

8. **Remark-** There is a typo mistake in legend of Fig 3, it should be KrasG12D instead of KrasG12D/Vav1

Response-

- Thanks for the remark, it was now corrected.

9. **Remark-** In Fig5 B and C, the size of the error bars of the quantification are not corresponding with the variation of intensity seen in the western blot images. Can authors provide explanation for that?

Response-

- The western blot (now Figure 5C) illustrates an experiment with 2 mice at 5-months post transgene induction, while the histogram represents multiple time points, several mice at each time point and between 3-4-time repetition on the western blot (detailed in Table S3). Additionally, the calculation represents the ratio between pErk/Erk. Taking into account, the number of repetitions and number of mice used, it is conceivable that error bars are so low.

10. **Remark-** Fig6B, the authors should provide the error bars for entire graph. Further, can authors provide the exact number of mice used instead of stating "several mice"?

Response-

- We now added the number of mice used in this experiment in Table S3. Error bars were added. In some of the columns in the histogram, the error bars are very small. Please see our response to your remark for point 9.

11. **Remark-** Fig 7D, error bars are absent from the graph.

Response-

- The error bar has been added to Fig. 7D.

12. **Remark-** In the last section of the Results (the text relating to Fig C&D), the data is referred to twice in the text, which is redundant.

Response-

- I apologize for our mistake. It has been now omitted.

13. **Remark-** In the Discussion section, there is a typo in last but one paragraph. It should be "raise" instead of "rase".

Response-

- I apologize for our mistake. It has been now corrected. Page 15, line 22.

Reviewer #2

1. **Remarks-**The authors state that "The removal of doxycycline from the drinking water of K-RasG12D/Vav1 mice (leading to a decrease in Vav1 transgene expression) elicited a marked decline in malignant lesions in the pancreas, further demonstrating that ADM generation requires the expression of Vav1 together with K-RasG12D". There is a conceptual flaw here: if vav1 was already expressed and gave rise to metaplastic lesions, decreasing its expression does not indicate vav1 requirement for ADM generation, but rather that vav1 is required for maintenance of ADM lesions and/or their progression to cancer. Please clarify. This is discussed to some extent already in the discussion; abstract has to reflect these points as well.

Response:

- Corrected in the Results section (page 10, lines 1, 2) and corrected in the abstract (page 2, line 10).

2. **Remarks-** Figure 7 shows the use of Azathioprine as a pharmacologic approach, though not entirely specific, to inhibit vav1 activity and demonstrate a phenotype of fewer lesions. Please clarify in abstract the model in which this was done- kras;vav1, not kras only mice. The implication is that vav1 activity (and not another cryptic activity of vav1) is needed for maintenance or further progression of lesions developed due to excess vav1 activity.

Response-

Added as justifiably pointed out in the abstract, page 2, lines 11-13.

3. **Remark-** Abstract lacks a concluding sentence- what is the bottom line regarding the role of vav1 in pdac?

Response-

- Added as requested to the abstract "These results suggest that Vav1 plays a role in the development of PDAC when co-expressed with K-Ras^{G12D} via its activity as a GEF for RacGTPase", page 2, lines 14, 15.

4. **Remarks-** If vav1 is required for ADM, it should be expressed in acinar cells, not just in full blown PDAC. Can the authors show this, in mouse and (preferably) in human material? Alternatively, its expression in acinar cells undergoing metaplasia can be transient - please either show data or discuss in text.

Response-

We agree with the reviewer and indeed Fig. 1 shows that at all time points, immunohistochemistry for GFP (expressed with the Vav1 transgene) showed

expression in many of the acinar cells. Also, Vav1 staining can be visualized in Fig. 3D, sections from K-Ras^{G12D}/Vav1 stained with anti-GFP Abs and sections of K-Ras^{G12D} stained with anti-Vav1 Abs (Fig. S3), in which Vav1 expression is demonstrated in acinar cells. We have clarified that in the text, **page 9, lines 21-23; page 8, line 3.**

Regarding human material - as Vav1 expression in human pancreas is likely induced by microenvironmental cues, it is not possible to assess this accurately.

5. **Remarks-** Is there evidence on whether the effects of vav1 are cell autonomous (as would be the null hypothesis) or whether non-transgenic cells are also affected? The idea that cytokines turn the cell autonomous nature of vav1 activity into a microenvironmental effect is very interesting.

Response-

- This point is extremely important and very interesting. We have expanded this section in the discussion, **page 15, lines 25-26, page 16, lines 1-14.**

Reviewer #3:

1) **Remarks-** The authors use frequently the term "oncogene" throughout the manuscript to mention/describe the Vav1 gene used in this report. To avoid confusion, since the constructs used here contain specifically the WT version of Vav1, they should correct this language.

Response-

- We have removed the term oncogene with reference to Vav1 as remarked by the reviewer throughout the manuscript.

2) **Remarks-** Figure 3: The authors need to provide stronger direct evidence to support/strengthen the conclusions they draw (page 9) from the current figure. I suggest, in particular, to (i) add WB assays showing the actual reduction of Vav1 under the conditions used (Dox removal 20 days prior to analysis), or (ii) at least, provide information on the half-life of the concerned cellular Vav1 proteins. Including a control panel of single K-Ras^{G12D} pancreas stained with anti-cytokeratine would also be advisable. Addressing these issues will also reinforce the conclusions drawn by the authors from the data in Figure 4.

Response-

- We have now added representative pictures of the pancreas of K-Ras^{G12D}/Vav1 mice (revised Figure 3D) with (+) and following removal of Dox (-) stained with GFP. It can be clearly noted that there is a marked reduction in GFP positive cells in mice in which Dox was removed from their drinking water.

- We included in the revised Figure 3A (left picture), a representative picture of a pancreatic section from K-Ras^{G12D} stained with anti-Cytokeratin Abs.

3) **Remarks-** Figure 5: A graph quantifying the pEGFR immunohistochemical signals detected in the different experimental groups (or direct WB data) would also be advisable here.

Response-

- We have now added sections of K-Ras^{G12D} and K-Ras^{G12D}/Vav1 stained with anti-EGFR that demonstrate the level of total EGFR in the sections (Fig. 5A, lower panel).

4) **Remarks-** Figure 6: The data shown in the three panels of this figure clearly support the synergism between Vav1 and K-ras resulting in significant increase of the levels of activated Rac1-GTP as indicated in page 11. However, the authors need to clarify experimental details of the data presented.

The Figure legend does not specify whether or not similar amounts of total protein were loaded in the WB assays in Fig 6 panel A. Is there a reduction of total Rac1 protein expressed upon co-expression of Vav1 and K-ras ? If so, the authors should mention this in the text. Similar questions arise when considering Figure 7: does azathioprine treatment cause increased expression levels of total Rac1 in comparison to untreated samples?

Response-

- As detailed in the revised manuscript, Material and Methods (**page 19, lines 6-14**), an equal amount of protein was loaded in the experiments. However, there are still differences which cannot be accurately controlled, especially since we use tissues and not cell-lines. Therefore, we also added a panel of actin loaded in each sample.
- Based on numerous western blots, we did not observe a reduction in total Rac1 expression in our experiments, including the ones presented in Fig. 7.

Remarks- The experimental details of the methodologies used for assays of Rac1 activation should be specified in Material and Methods. The authors seem to base all their determinations exclusively on WB assays using antibodies against Ras1-GTP, however other authors use immunoprecipitation and subsequent WB. The positive and negative controls for the immunofluorescence assays shown in panel C should also be shown, or at least mentioned.

- First, as suggested, we have added information regarding the detection of Rac1-GTP activation in the Material and Methods section (**page 18, lines 7-17**). Second, we have used anti-Rac1-GTP antibodies and not anti-Ras1-GTP antibodies. Third, there is no indication that using first immunoprecipitation and then western blotting with anti-Rac1 Abs is a preferable approach to the one we used. Moreover, we present here data of immunofluorescence with anti-Rac1-GTP which further strengthens our results. Fourth, Fig. 6C demonstrates negative biological controls. Also, an additional control such as only secondary Abs is now mentioned in the figure legend (**page 29, lines 4-7**). This control was negative and therefore not mentioned in the previous version. The antibodies used are specified in Table S2.

5) **Remarks-** Regarding mechanistic aspects:

- The authors suggest (p. 16) the possibility that Vav1 synergizes with oncogenic K-ras through its participation in chemokine and cytokine signaling. This view could be reinforced if the authors could include in this paper some data regarding the status of some inflammation biomarkers in the different experimental groups analyzed here.

- A Pubmed search on this subject and authors brings out an old publication (Katzav et al, Oncogene, 1995) entitled "Vav and Ras induce fibroblast transformation by overlapping signaling pathways which require c-myc function". In view of the many publications on pancreatic cancer and K-ras-mediated transformation that have been published in more recent years, the authors may consider discussing here whether the synergism proposed for Vav1 and K-ras may also require c-myc contribution.

Response-

- I agree with the reviewer that the question of how is the microenvironment affected by Vav1 expression and whether it affects cytokine release is highly interesting. Please see our response to reviewer 2 above.
- Thanks to reviewers' suggestion, we have now included a paragraph relating to c-Myc/K-Ras/Vav1 as a possible downstream pathway in PDAC generation, **page 16, lines 15-26, page 17, lines 1,2.**

Minor points

1) **Remarks-** Please define units for panel B Figure EV 3

Response-

- According to the instructions of the journal, Figure EV 3 is Fig. S3 in the revised manuscript.
- The Y axis in Fig. S3B, refers to the mean values of Vav1-positive acinar/ductal cells at each time point indicated. Endogenous Vav1 expression was quantified by counting Vav1-positive acinar and ductal cells in K-Ras^{G12D} mice at different times after transgene induction. The number of mice used in this experiment is specified at the legend to this figure (page 30, lines 23-26, page 31, lines 1,2).. Five randomly chosen fields were counted in mouse sections. We have now changed to Y axis to: No. Cells expressing Vav1.

2) **Remarks-** Please correct typos (i.e: co-epxressed, p.8; immunohistochemistry, p.10; signaling, p.14; Remarkably, p.12; rases, p.16; ...)

Response-

- **Corrected in the revised manuscript.**

3) **Remarks-** Add parameters of statistical significance missing in some graphs (i.e.: Figure 5).

Response-

Corrected in the revised figure.

4) **Remarks-** Number of mice used for each experiment must be indicated in all figure legends.

Response-

Number of mice used in experiments presented in figures 1, 4, 5, and 6 are presented in Table S3. Number of mice used in the experiment presented in Figures 2, 3 and 7 are documented in the legends to these figures.

We believe that the revised manuscript has improved thanks to the reviewers' comments and hope that the revised manuscript will now be accepted for publication in Life science Alliance.

Sincerely yours,

Shulamit Katzav Shapira, Professor.
Developmental Biology & Cancer Research
IMRIC
Faculty of Medicine-The Hebrew University

March 18, 2020

RE: Life Science Alliance Manuscript #LSA-2020-00661-TR

Prof. Shulamit Katzav
Faculty of Medicine-The Hebrew University
Developmental Biology & Cancer Research
P.O.Box 12272
Jerusalem 91120
Israel

Dear Dr. Katzav,

Thank you for submitting your revised manuscript entitled "Vav1 and Mutant K-Ras Synergize in the Early Development of Pancreatic Ductal Adenocarcinoma in Mice". I assessed your response to the reviewer concerns and the introduced changes, and I appreciate how you dealt with the initial concerns. I would thus be happy to publish your paper in Life Science Alliance pending final revisions necessary to meet our formatting guidelines:

- Please upload a new Figure 6 - the source data do not match the data shown for anti-Rac1-GTP and anti-Rac1 (Control and Vav1 lanes show the signal of V1 and V2 instead of the signal for Control and V1)
- Please define what kind of error bars are shown when stating „Error bars are shown" in the figure legends
- Please change the callout to Fig 5A, B on page 10 to "Fig 5A,B" (currently the word "Fig" is missing)
- Please add scale bars to Fig. 6C & Fig. S1C and increase the scale bars already present on other panels (too difficult to appreciate)

A. FINAL FILES:

B. MANUSCRIPT ORGANIZATION AND FORMATTING:

Sincerely,

Dear Dr. Leibfried,

Thank you for your e-mail concerning our revised manuscript #LSA-2020-00661-T of February 19th, 2020, and the opportunity to resubmit a revised version of our manuscript.

As for the concerns that you raised:

- Editor- Please upload a new Figure 6 - the source data do not match the data shown for anti-Rac1-GTP and anti-Rac1 (Control and Vav1 lanes show the signal of V1 and V2 instead of the signal for Control and V1)
Response- I revised the figure. However, the order of the lanes is: N, V2, R3, R1, R/V3, R/V1 as I wrote in my correspondence to Reilly Lorenz
- Editor- Please define what kind of error bars are shown when stating „Error bars are shown" in the figure legends
Response- “Error bars are shown” refers to standard error of the mean (SEM). We have now corrected it in the corresponding figure legends, as follows: Fig. 2B, Fig. 3C, Fig. 4B, Fig. 5C, Fig. 6B, Fig. 7B, Fig. 7C, Fig. S3B and Fig. S4B.
- Editor- Please change the callout to Fig 5A, B on page 10 to "Fig 5A,B" (currently the word "Fig" is missing)
Response- Corrected
- Editor- Please add scale bars to Fig. 6C & Fig. S1C and increase the scale bars already present on other panels (too difficult to appreciate)
Response-
 - Unfortunately, the immunofluorescence pictures shown in Fig. 6C & Fig. S1C were taken a while ago using microscopes that did not have an inherent capability to include scale bars. If permitted, we can add estimated scale bars, based on comparison to newer pictures. Otherwise we will leave the text in the legend stating the original magnification.

- We have increased the scale bars in all the other IHC pictures, so that the scale bars are now more visible. The revised figures are uploaded.

Please let me know if additional changes are required.

Sincerely yours,

Shulamit Katzav Shapira, Professor.
Developmental Biology & Cancer Research
IMRIC
Faculty of Medicine-The Hebrew University
P.O.Box 12272
Jerusalem 91120
Israel

Tel: 972-2-6758350; Fax: 972-2-6757482
E-Mail address: shulamitk@ekmd.huji.ac.il

March 23, 2020

RE: Life Science Alliance Manuscript #LSA-2020-00661-TRR

Prof. Shulamit Katzav
Faculty of Medicine-The Hebrew University
Developmental Biology & Cancer Research
P.O.Box 12272
Jerusalem 91120
Israel

Dear Dr. Katzav,

Thank you for submitting your Research Article entitled "Vav1 and Mutant K-Ras Synergize in the Early Development of Pancreatic Ductal Adenocarcinoma in Mice". It is a pleasure to let you know that your manuscript is now accepted for publication in Life Science Alliance. Congratulations on this interesting work.

DISTRIBUTION OF MATERIALS:

Again, congratulations on a very nice paper. I hope you found the review process to be constructive and are pleased with how the manuscript was handled editorially. We look forward to future exciting submissions from your lab.

Sincerely,
